# Long-Term Follow-Up of Bridging Therapies Prior to CAR T-Cell Therapy for Relapsed/Refractory Large B Cell Lymphoma

**DOI:** 10.3390/cancers15061747

**Published:** 2023-03-14

**Authors:** Colton Ladbury, Savita Dandapani, Claire Hao, Mildred Fabros, Arya Amini, Sagus Sampath, Scott Glaser, Karen Sokolov, Jekwon Yeh, John H. Baird, Swetha Kambhampati Thiruvengadam, Alex Herrera, Matthew Mei, Liana Nikolaenko, Geoffrey Shouse, Lihua E. Budde

**Affiliations:** 1Department of Radiation Oncology, City of Hope National Medical Center, Duarte, CA 91010, USA; cladbury@coh.org (C.L.); sdandapani@coh.org (S.D.); clhao@coh.org (C.H.); mfabros@coh.org (M.F.); aamini@coh.org (A.A.); ssampath@coh.org (S.S.); sglaser@coh.org (S.G.); ksokolov@coh.org (K.S.); jeyeh@coh.org (J.Y.); 2Department of Hematology and Hematopoietic Cell Transplantation, City of Hope National Medical Center, Duarte, CA 91010, USA; jbaird@coh.org (J.H.B.); skambhampati@coh.org (S.K.T.); aherrera@coh.org (A.H.); mamei@coh.org (M.M.); liananik@yahoo.com (L.N.); gshouse@coh.org (G.S.)

**Keywords:** bridging therapy, bridging radiation, large B-cell lymphoma (LBCL), chimeric antigen receptor T-cell (CAR T)

## Abstract

**Simple Summary:**

Bridging therapy (BT) in the form of systemic therapy (ST), radiation therapy (RT), or combined-modality therapy (CMT) is increasingly being utilized prior to chimeric antigen receptor (CAR) T-cell therapy for large B-cell lymphoma (LBCL). Prior institutional experiences with BT, and RT in particular, suggest it does not lead to increased toxicity or compromise outcomes. Further investigation of the optimal use of BT is warranted. In this research, we sought to evaluate the impact of BT in a large institutional cohort of patients who received commercial CAR T-cell therapy for LBCL, specifically examining the effect of BT modality and disease burden on outcomes. Here, we report the long-term outcomes of our CAR T cohort. The patients with limited disease treated with RT had favorable outcomes. Prospective studies are warranted to better characterize the optimal management of patients with relapsed/refractory LBCL who are planning to undergo CAR T-cell therapy to optimize treatment.

**Abstract:**

Background: Bridging therapy (BT) with systemic therapy (ST), radiation therapy (RT), or combined-modality therapy (CMT) is increasingly being utilized prior to chimeric antigen receptor (CAR) T-cell therapy for large B-cell lymphoma (LBCL). We report the long-term outcomes of the patients who received commercial CAR T-cell therapy with or without BT. Methods: The patients with LBCL who underwent infusion of a commercial CD19 CAR T product were eligible. The radiation was stratified as comprehensive or focal. The efficacy outcomes and toxicity were analyzed. Results: In total, 156 patients were included and, of them, 52.5% of the patients received BT. The median progression-free survival (PFS) was 0.65 years in the BT cohort compared to 1.45 years in the non-BT cohort. The median overall survival (OS) was 3.16 years in the BT cohort and was not reached in the non-BT cohort. The patients who received comprehensive radiation (versus focal) had significantly improved PFS and OS, achieving a 1-year PFS of 100% vs. 9.1% and 1-year OS of 100% vs. 45.5%. There was no difference in the severe toxicity between any of the nonbridging or BT cohorts. Conclusions: BT did not appear to compromise outcomes with respect to response rates, disease control, survival, and toxicity. The patients with limited disease treated with RT had favorable outcomes.

## 1. Introduction

Aggressive large B-cell lymphomas (LBCL), including diffuse LBCL (DLBCL), primary mediastinal LBCL (PMLBCL), and transformed follicular lymphoma (tFL), represent the most common form of non-Hodgkin lymphoma (NHL), accounting for approximately one-third of all NHL cases [1]. Among patients with LBCL, approximately one-third of patients will either have primary refractory disease or will relapse after initial chemotherapy [2]. Historically, patients with relapsed and refractory DLBCL (R/R DLBCL) have poor prognoses [3,4,5]. Chimeric antigen receptor T-cell (CAR T) therapy has become an important treatment option for these patients, with three Food and Drug Administration-approved autologous CD19-directed therapies (axicabtagene ciloleucel [axi-cel], lisocabtagene maraleucel [liso-cel], and tisagenlecleucel [tisa-cel]) being available, all of which demonstrate high response rates and durable responses in a significant subset of patients [6,7,8].

Though based on comparisons to historical data, CAR T appears to be superior to chemoimmunotherapy-based salvage; a significant limitation intrinsic to autologous CAR T products is the interval between apheresis and CAR T infusion required for product manufacturing, which approximately averages at one month at our institution. Given the aggressive nature of the underlying disease, many patients require some type of bridging therapy (BT) to control their disease during this time. BT can include steroids, chemotherapy, targeted therapy, or radiation therapy (RT). Notably, in an analysis by the United States Lymphoma CAR T Consortium, BT was associated with inferior overall survival (OS) on multivariate analysis, although a specific contribution by BT modality was not provided [9]. Overall, prior institutional experiences with BT have also supported this finding, although patients who received radiation BT appeared to have comparable outcomes to patients who did not receive BT [10,11,12,13,14].

Given the existing data, further investigation of the optimal use of BT is warranted. In this study, we sought to evaluate the impact of BT in a large institutional cohort of patients who received commercial CAR T-cell therapy for LBCL, specifically examining the effect of BT modality and disease burden on outcomes. Here, we report the long-term outcomes of our CAR T cohort.

## 2. Materials and Methods

### 2.1. Patient Selection

After institutional review board approval, we conducted a retrospective analysis of patients with R/R LBCL who underwent leukapheresis and infusion of a commercially available CD19 CAR T product (either axi-cel or tisa-cel) between December 2017 and March 2021. During this time period, consecutive patients were prospectively documented in an institutional database, which was queried for this analysis. Liso-cel was approved in February 2021 and, therefore, no patients that received liso-cel were included. Patients with DLBCL, PMBCL, and tFL were included.

### 2.2. Bridging Therapy and CAR T Infusion

BT was defined as systemic therapy (ST; including steroids, chemotherapy, and/or targeted therapy), radiation therapy (RT), or combined-modality therapy (CMT; both ST and RT) received prior to lymphodepleting chemotherapy and within 60 days of CAR T infusion. ST was administered at the discretion of the treating hematologist. The RT treatment site and dose were administered at the discretion of the treating radiation oncologist. The patients who received low-dose steroids for side-effect management during radiation were not considered to have received CMT. The radiation that encompassed all metabolically active tumors at the time of simulation was considered comprehensive, while radiation that only targeted certain sites of the disease was considered focal. Comprehensive RT was administered whenever it was felt to be feasible to encompass all the disease in the radiation field without excessive toxicity. Lymphodepletion chemotherapy consisted of cyclophosphamide (500 mg/m^2^) and fludarabine (30 mg/m^2^) administered on days -5, -4, and -3. CAR T-cell treatment consisted of a single infusion of axi-cel or tisa-cel on day 0.

### 2.3. Toxicity Evaluation

The toxicity data were prospectively gathered. The consensus criteria were used for grading the cytokine release syndrome (CRS) and immune effector cell-associated neurotoxicity syndrome (ICANS) severity according to the CARTOX criteria [15] until April 2019 and thereafter according to the American Society for Transplantation and Cellular Therapy criteria [16].

### 2.4. Risk Factor Definition

The international prognostic index was calculated based on clinical factors at the time of leukapheresis [17]. High-grade lymphoma (double or triple hit) was identified using fluorescence in situ hybridization (FISH); bulky disease was defined as ≥7.5 cm. The number of disease sites and SUV_max_ were determined based on the most recent pre-CAR T positron emission tomography–computed tomography (PET-CT) performed prior to BT.

### 2.5. Study end Points and Statistical Methods

The disease response evaluations were conducted using PET-CT, based on the Lugano classification [18]. Progression and relapse were defined using clinical, radiographic, or pathologic confirmation. All the survival measures were defined from the date of the CAR T-cell infusion. The progression-free survival (PFS) was defined as any disease progression, relapse, or death resulting from any cause. OS was defined as death resulting from any cause.

The Kaplan–Meier method was used for survival time estimations [19]. The differences between the groups were compared using the log–rank test. The median follow-up times and 95% confidence intervals (CIs) were calculated using the reverse Kaplan–Meier method. A univariate and multivariable logistic regression was performed to test the associations between the predictive factors and treatment toxicity. The multivariable models were performed using a backwards selection method. The cutoff for data analysis was 17 June 2022. Between the patient cohorts, the categorical variables were compared using Fisher’s exact test and the continuous variables were compared using the Student t-test. All the comparisons were two-sided and used a statistical significance threshold of *p* < 0.05. The statistical analyses were performed using Python 3.8 (PSF, Wilmington, DE, USA).

## 3. Results

### 3.1. Patient Details

A total of 156 patients ultimately received infusions of axi-cel (n = 153) or tisa-cel (n = 3) (Table 1). The median age of the entire cohort was 62 years (range: 21–84 years) and 62.2% of the patients were male. Most of the patients were diagnosed with DLBCL (68.6%). The patients received a median of two prior lines of treatment (range: two–eight). Of the patients who underwent CAR T infusion, at the time of treatment, 15.4% had MYC and BCL-2 and/or BCL-6 rearrangements, consistent with double-hit/triple-hit lymphoma, 3.8% had an Eastern Cooperative Oncology Group (ECOG) performance score of from 2 to 3, 25.6% had bulky disease, 66.7% had extranodal disease, 41.7% had an elevated LDH, and 10.3% had an IPI of ≥3. 

Overall, 82 (52.5%) of the patients who received a CAR T infusion received some form of BT, with 63 (76.8%) receiving ST, 12 (14.6%) receiving RT, and 7 (8.5%) receiving CMT. The median time between leukapheresis and CAR T infusion was 27 days (range: 20–242 days), and there was no difference between the patients who received BT and those who did not (median: 27 vs. 27 days; *p* = 0.730). In one patient, there was a 242-day interval between leukapheresis and CAR T infusion due to development of an active hepatitis B infection that required treatment before infusion. The patients who received BT were more likely to have an IPI of ≥3 (17.1 vs. 2.7%; *p* = 0.007), to have an elevated LDH (53.7 vs. 28.4%; *p* = 0.006), to have a higher median SUV_max_ (16.9 vs. 14.8; *p* < 0.001), and have a larger maximum tumor dimension (54 vs. 38 mm; *p* < 0.001) when compared to the patients who did not receive BT. There was no significant difference in the prior lines of treatment (median: 2 vs. 3; *p* = 0.283).

Within the bridging cohorts, there were differences in the baseline ECOG score, extranodal disease, LDH, maximum disease size, and SUV_max_. A total of 60.3%, 58.3%, and 71.4% of patients in the ST, RT, and CMT cohorts were older than 60 years, respectively. There was a significant difference in time between leukapheresis and CAR T infusion in the RT and ST bridging cohorts (27 days [range: 24–242 days] vs. 27 days [range: 20–57 days]; *p* = 0.023).

Within the ST bridging group, 53 patients (84.1%) received cytotoxic chemotherapy, 6 patients (9.5%) received targeted therapies, and 4 (6.3%) patients received corticosteroids. Among the patients who received ST, the most common regimen was rituximab, bendamustine, and polatuzumab vedotin (n = 27).

In the RT cohort, the median RT dose was 20 Gy (range: 4–40 Gy), with a median fraction size of 2.25 Gy (range: 2–3 Gy). A total of eight patients (66.7%) received comprehensive RT to all sites of metabolically active disease while four (33.3%) received focal RT. RT was initiated prior to leukapheresis in seven patients (58.3%). The patient-level information on the individuals who received RT BT is available in Table 2.

In the CMT cohort, five patients (83.3%) received cytotoxic chemotherapy with rituximab, bendamustine, and polatuzumab vedotin and one patient (16.7%) received targeted therapy with pembrolizumab. The median RT dose was 20 Gy (range: 14–30 Gy), with a median fraction size of 2.86 Gy (range: 1.4–4 Gy). All the patients (100%) received focal RT. RT was initiated prior to leukapheresis in three patients (50%). The patient-level information on the individuals who received CMT BT is available in Table 2.

### 3.2. Hemato-Oncologic Outcomes

The median follow-up time in our study was 1.60 years (range: 0.02–4.01 years) and 2.05 years (range: 1.00–4.01 years) in the living patients. At the time of analysis, 93 of the 156 patients (59.6%) had progressed or died. The median PFS and OS of the entire cohort was 1.05 years (95% CI: 0.55–1.95 years) and 3.16 years (95% CI: 2.09—Not Reached [NR] years), respectively. The 1-year PFS and OS of the entire cohort was 51.9% (95% CI: 43.8–59.4 %) and 75.0% (95% CI: 67.4–81.1%), respectively. The overall response rate (ORR) was 77.6% in the entire cohort, with 65.2% achieving a complete response (CR).

There was no significant difference in the PFS (*p* = 0.160) or OS (*p* = 0.158) between the BT and non-BT cohorts (Figure 1A,B). The median PFS was 0.65 years (95% CI: 0.28–1.79 years) in the BT cohort compared to 1.45 years (95% CI: 0.58—NR years) in the non-BT cohort. The median OS was 3.16 years (95% CI: 1.37—NR years) in the BT cohort but was not reached (NR) in the non-BT cohort (95% CI: 2.33—NR years). The 1-year PFS was 45.1% (95% CI: 34.2–55.5%) in the BT cohort compared to 59.5% (95% CI: 47.4–69.6%) in the non-BT cohort. The 1-year OS was 69.5% (95% CI: 58.3–78.3%) in the BT cohort compared to 81.1% (95% CI: 70.2–88.3%) in the non-BT cohort. There was no difference in the ORR (90.3 vs. 89.0%; *p* = 0.759) or CR rates (61.0 vs. 68.9%; *p* = 0.322) in the patients who did or did not receive BT (Table 3).

The patients who received RT numerically had the greatest PFS and OS, with a 1-year PFS and OS of 75.0% (95% CI: 40.8–91.2%) and 91.7% (95% CI: 53.9–98.8%), respectively. The median PFS and OS in the RT BT cohort were 3.32 years (95% CI: 0.13–3.32 years) and 3.32 years (95% CI: 1.17–3.32 years). With a 1-year PFS of 0% (95% CI: 0–0%) and a median PFS of 0.24 years (95% CI: 0.02–0.45 years), the patients who received CMT BT had inferior PFS to the patients who received no BT (*p* < 0.001), ST BT (*p* = 0.004), or RT BT (*p* = 0.001).

Among the patients who received radiation, 7 (7 RT) received comprehensive radiation to all sites of a metabolically active tumor while 11 (4 RT, 7 CMT) received focal radiation. The characteristics of these patients are summarized in Table 4. The patients who received comprehensive radiation had a significantly improved PFS relative to those who received focal radiation (*p* < 0.001), with a 1-year PFS of 100% (95% CI: 100–100%) vs. 9.1% (95% CI: 0.5–33.3%) (Figure 2). The OS was also significantly better in the patients treated with comprehensive RT relative to those who received focal radiation (*p* = 0.010), with a 1-year OS of 100% (95% CI: 100–100%) vs. 45.5% (95% CI: 16.7–70.7%). There was no significant difference in the ORR (54.5% vs. 87.5; *p* = 0.305) but the CR was significantly improved in the comprehensive RT cohort (27.3 vs. 87.5%; *p* = 0.033). Between the groups, the patients treated with focal RT BT had a significantly greater median maximum lesion size (66 mm vs. 49 mm; *p* < 0.001) and median number of disease sites (7 vs. 2; *p* = 0.002).

### 3.3. Treatment Related Toxicities

There was no difference in severe (grade ≥ 3) cytokine release syndrome (CRS) or immune effector cell-associated neurotoxicity syndrome (ICANS) between any of the nonbridging or BT cohorts, though there was significantly more any grade CRS in the BT cohort (91.5% vs. 78.4%; *p* = 0.038) (Table 4). On the multivariable logistic regression, prior lines of therapy, bridging therapy, and having more than one disease site were associated with the development of any grade CRS, while having a LDH >2x ULN and total number of disease sites were associated with severe CRS (Table 5). Only CRS was associated with any grade ICANS, while having an LDH >2xULN and total disease sites were associated with severe ICANS. There was no significant difference in tocilizumab administration, steroid use, or ICU admission between the BT and non-BT cohorts. There was one death due to toxicity (septic shock [n = 1]) in a patient who did not receive BT and two deaths due to toxicity (septic shock [n = 1] and neurotoxicity [n = 1]) in the patients who received ST. There were no deaths resulting from toxicity in either the RT or CMT BT cohorts.

## 4. Discussion

We report on a large institutional experience of patients with R/R LBCL who underwent treatment with commercially available anti-CD19 CAR T-cell therapy. To our knowledge, this is the largest single institution analysis, with the longest follow-up (median of 2 years in living patients compared to 0.9 years in the other largest series [11]) evaluating the effect of BT on outcomes post CAR T. In contrast to other published series, the patients who received BT did not have inferior OS or PFS that reached statistical significance. This could be attributed to referral patterns at our institution for RT BT, where the threshold for referral is lower and includes bulky or persistent disease that may not be symptomatic or life-threatening, which is likely consistent with more favorable prognoses. This is supported by the fact that our patient characteristics are otherwise very similar to those reported by Pinnix et al. [11]. Consistent with other series, the patients who received RT BT, particularly comprehensive RT BT where all sites of disease could be treated, had favorable outcomes. Overall, this study adds to the growing body in the literature that, in selected patients, BT can help control disease burden prior to CAR T-cell infusion.

Prior evidence has suggested that, when possible, encompassing all active disease in the RT BT fields can achieve favorable results. In an analysis by Pinnix et al., the 1-year OS and PFS were 71% and 57%, respectively, in the patients treated with comprehensive RT BT, compared to 28% and 17% in the patients treated with focal RT BT. Although there was not a statistically significant difference, there was a strong numerical trend. Our study found the same trend but, in our cohort, this reached statistical significance, as both OS (*p* = 0.001) and PFS (*p* < 0.001) were significantly improved with comprehensive RT BT compared to focal RT BT. Certainly, in both studies this may be driven by selection bias; the patients with more limited disease or lower overall tumor burden, who already would have better prognoses, are also most amenable to comprehensive RT BT. Therefore, this finding is expected and represents significant selection bias. Nonetheless, treating all gross diseases should be considered whenever feasible as a means of improving local control, as illustrated by the favorable outcomes when examining the comprehensive RT cohort in isolation.

Of interest, the radiation dose used in this cohort was on average lower than what was used in previous studies, suggesting that doses normally required for definitive tumor ablation may be avoided as bridging prior to CAR T therapy, without a loss of efficacy. This is consistent with preclinical work, where low dose radiation conditioning was shown to mitigate the antigen-negative tumor relapse and potentiate CAR T-cell efficacy via TRAIL-mediated apoptotic death [20]. The existing case reports also indicate that disease progression after CAR T-cell therapy only occurred in areas that harbored disease before infusion but were not included in the radiation field [11]. While CAR T cells do not rely on antigen processing and presentation, and it is unclear if radiation may have an immunogenic, immunosuppressive, or irrelevant effect on CAR T cell therapy, a current theory is that a subtherapeutic RT dose might promote the migration of CAR T cells to the tumor site and increase effector functions [21]. Therefore, the findings of our study support that low-dose radiation versus high-dose radiation may activate the immune system differently. Future preclinical studies are necessary to understand how radiation can augment the CAR T-cell response.

In contrast, a retrospective analysis of the patterns of failure following RT BT showed a clear dose response where no local failures occurred at doses higher than 37.5 Gy or an equivalent dose in 2 Gy fractions (EQD2) of 39 Gy [22]. Of the lesions that received less than this dose (median EQD2 of entire cohort was 24 Gy), nine (21.4%) experienced in-field failures. The risk of failure appeared to be associated with the lesion size as seven out of nine failures had a metabolic tumor volume >50 cc. Another analysis of the patterns of failure in R/R DLBCL following CAR T-cell therapy similarly reported that the lesions that were ≥5 cm, had SUV_max_ ≥ 10, or were extranodal were associated with a greater risk of local failure, with the larger lesions also being associated with worse OS [23]. Notably, 36% of the patients had purely local progression, while 86% of the patients had some component of local progression. Our work in combination with these patterns of failure studies suggest that the dose might be tailored based on the disease burden characteristics; large lesions or lesions with high metabolic activity may require higher radiation doses while smaller less-active lesions might be amenable to lower doses. Optimizing and minimizing radiation doses will be an area of ongoing research to limit toxicity but still achieve disease control.

Importantly, as with prior studies, there was no indication that bridging the therapy of any type was associated with an increased incidence of toxicity or complications following CAR T-cell infusion. Indeed, across all subgroups, there was no difference regarding severe CRS, severe ICANS, corticosteroid use, tocilizumab administration, or ICU admission. In the multivariable analysis, BT was associated with higher rates of any grade CRS, although this trend did not extend to grade 3+ CRS. The significance of this finding is unclear, however, given that the overall disease burden is a known risk factor for both requiring BT and the development of CRS [24]. Although metabolic tumor volume estimates are not available in our cohort, the median size of the largest lesion, elevation in LDH, and median SUV_max_ were all higher in the BT cohort, suggesting that these patients may be predisposed to developing CRS even without BT. The important next steps will be to ascertain whether post-BT and pre-CAR T-cell infusion disease burden influences the risk of toxicity to determine if debulking might yield improvements [25]. In the multivariable analysis, BT was not associated with ICANS, either grade ≥1 or grade ≥3, though having experienced CRS was associated with Grade ≥1 ICANS, which is consistent with the other series.

Although our results suggests that BT is safe and effective, our study does have several limitations. First, given the retrospective nature of the study, BT was administered at physician discretion and, therefore, was not standardized. Additionally, only a small number of patients received treatment with any given modality. As such, no conclusions can be drawn regarding the optimal timing and dosing of BT based on these data. Even so, our study is novel in how a lower radiation dose was used in most patients, which appeared to result in comparable CAR T outcomes, some of which have remained durable with extended follow-up. The results of this study are similar to other institutional experiences and argue for a closer examination of the synergy between RT and CAR T-cell therapy. Larger multi-institutional and prospective studies will provide further insight into how RT BT might best be used in combination with CAR T to improve treatment outcomes.

## 5. Conclusions

In our cohort of patients with R/R DLBCL who underwent CAR T-cell infusion, we found that BT did not compromise outcomes with respect to response rates, disease control, survival, and toxicity. The patients with limited disease treated with RT had favorable outcomes. Prospective studies are warranted to better characterize the optimal management of patients with R/R LBCL who are planning to undergo CAR T-cell therapy to optimize treatment.

## Figures and Tables

**Figure 1 cancers-15-01747-f001:**
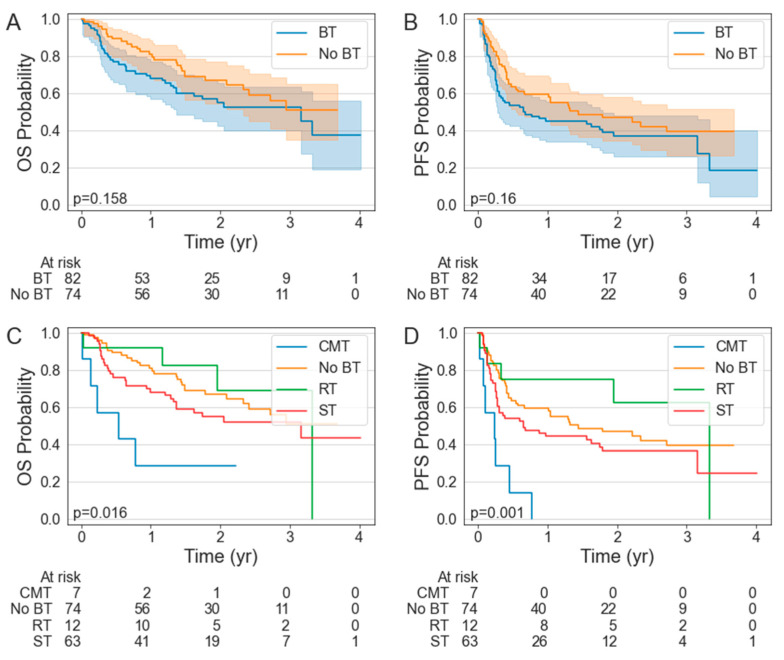
Kaplan–Meier estimates of overall survival (**A**,**C**) and progression-free survival (**B**,**D**) stratified by bridging therapy status.

**Figure 2 cancers-15-01747-f002:**
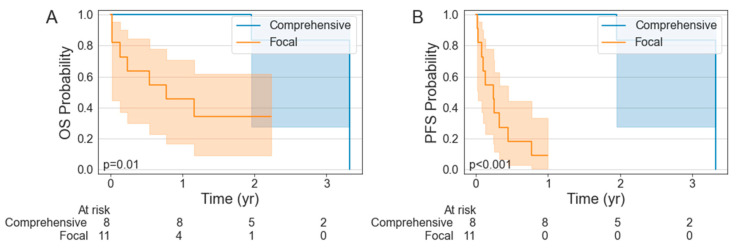
Kaplan–Meier estimates of overall survival (**A**) and progression-free survival (**B**) stratified by receipt of comprehensive versus focal bridging radiotherapy.

**Table 1 cancers-15-01747-t001:** Patient and treatment characteristics of patients who underwent CAR T-cell therapy according to type of bridging therapy.

			BT				*p*						
Characteristic	All Patients (n = 156)	No BT (n = 74)	BT (n = 82)	ST (n = 63)	CMT (n = 7)	RT (n = 12)	No BT vs. BT	ST vs. No BT	ST vs. CMT	ST vs. RT	No BT vs. CMT	No BT vs. RT	CMT vs. RT
Age (median [range])	62.0 (21–84)	61.0 (22–80)	62.5 (21–84)	62.0 (23–84)	64.0 (25–71)	65.0 (21–75)	0.896	0.856	0.744	0.965	0.798	0.889	0.819
>60	88 (56.4%)	38 (51.4%)	50 (61.0%)	38 (60.3%)	5 (71.4%)	7 (58.3%)	0.294	0.379	0.87	1	0.534	0.891	0.938
Male	97 (62.2%)	47 (63.5%)	50 (61.0%)	41 (65.1%)	4 (57.1%)	5 (41.7%)	0.872	0.991	1	0.229	1	0.264	0.861
Race and Ethnicity							0.121	0.137	0.264	0.675	0.637	0.052	0.086
Asian	35 (22.4%)	10 (13.5%)	25 (30.5%)	19 (30.2%)	0 (0.0%)	6 (50.0%)							
Black	8 (5.1%)	4 (5.4%)	4 (4.9%)	3 (4.8%)	1 (14.3%)	0 (0.0%)							
Hispanic White	38 (24.4%)	19 (25.7%)	19 (23.2%)	16 (25.4%)	1 (14.3%)	2 (16.7%)							
Native Hawaiian or other Pacific Islander	1 (0.6%)	0 (0.0%)	1 (1.2%)	1 (1.6%)	0 (0.0%)	0 (0.0%)							
Non-Hispanic White	73 (46.8%)	40 (54.1%)	33 (40.2%)	24 (38.1%)	5 (71.4%)	4 (33.3%)							
Unknown	1 (0.6%)	1 (1.4%)	0 (0.0%)	0 (0.0%)	0 (0.0%)	0 (0.0%)							
ECOG PS 2–3	6 (3.8%)	1 (1.4%)	5 (6.1%)	1 (1.6%)	1 (14.3%)	3 (25.0%)	0.262	1	0.473	0.009	0.404	0.004	1
Stage III/IV	109 (69.9%)	46 (62.2%)	63 (76.8%)	49 (77.8%)	7 (100.0%)	7 (58.3%)	0.069	0.073	0.37	0.29	0.11	1	0.147
Extranodal disease	104 (66.7%)	47 (63.5%)	57 (69.5%)	43 (68.3%)	7 (100.0%)	7 (58.3%)	0.533	0.688	0.186	0.738	0.124	0.982	0.147
>1 Site Extranodal disease	81 (51.9%)	39 (52.7%)	42 (51.2%)	31 (49.2%)	7 (100.0%)	4 (33.3%)	0.98	0.813	0.031	0.487	0.044	0.351	0.018
IPI							0.02	0.021	0.619	0.154	0.013	0.126	0.316
IPI ≥ 3	16 (10.3%)	2 (2.7%)	14 (17.1%)	10 (15.9%)	2 (28.6%)	2 (16.7%)	0.007	0.016	0.751	1	0.035	0.164	0.976
Pathology							0.202	0.439	0.891	0.287	0.73	0.149	0.607
DLBCL	107 (68.6%)	47 (63.5%)	60 (73.2%)	44 (69.8%)	5 (71.4%)	11 (91.7%)							
PMBCL	8 (5.1%)	6 (8.1%)	2 (2.4%)	2 (3.2%)	0 (0.0%)	0 (0.0%)							
tFL	41 (26.3%)	21 (28.4%)	20 (24.4%)	17 (27.0%)	2 (28.6%)	1 (8.3%)							
Cell of Origin							0.678	0.797	0.727	0.285	0.738	0.278	0.917
ABC	54 (34.6%)	23 (31.1%)	31 (37.8%)	22 (34.9%)	3 (42.9%)	6 (50.0%)							
GCB	86 (55.1%)	43 (58.1%)	43 (52.4%)	36 (57.1%)	3 (42.9%)	4 (33.3%)							
HGBL-DH/TH	24 (15.4%)	9 (12.2%)	15 (18.3%)	11 (17.5%)	3 (42.9%)	1 (8.3%)	0.402	0.527	0.273	0.718	0.103	1	0.231
Double Expresser	26 (16.7%)	14 (18.9%)	12 (14.6%)	7 (11.1%)	1 (14.3%)	4 (33.3%)	0.616	0.305	1	0.121	1	0.45	0.712
Max Dimension (mm; median [range])	50.0 (5.0–212.0)	38.0 (5.0–122.0)	54.0 (10.0–212.0)	55.0 (10.0–212.0)	69.0 (14.0–137.0)	52.0 (13.0–122.0)	<0.001	<0.001	<0.001	0.55	<0.001	<0.001	<0.001
Bulky Disease (≥7.5 cm)	40 (25.6%)	16 (21.6%)	24 (29.3%)	18 (28.6%)	4 (57.1%)	2 (16.7%)	0.364	0.459	0.265	0.618	0.104	0.993	0.187
Large Disease (≥5 cm)	75 (48.1%)	28 (37.8%)	47 (57.3%)	37 (58.7%)	3 (42.9%)	7 (58.3%)	0.023	0.023	0.687	1	1	0.306	0.861
LDH							0.006	0.002	0.21	0.542	0.596	0.648	0.645
>ULN and <2xULN	53 (34.0%)	17 (23.0%)	36 (43.9%)	31 (49.2%)	1 (14.3%)	4 (33.3%)							
LDH > 2xULN	12 (7.7%)	4 (5.4%)	8 (9.8%)	6 (9.5%)	1 (14.3%)	1 (8.3%)	0.473	0.552	1	1	0.911	1	1
Maximum SUV (median [range])	16.0 (1.4–56.6)	14.8 (2.4–56.6)	16.9 (1.4–40.0)	17.0 (1.4–40.0)	15.7 (12.8–21.6)	17.4 (4.8–34.8)	<0.001	<0.001	0.711	0.576	<0.001	<0.001	0.477
Maximum SUV > 10	118 (75.6%)	52 (70.3%)	66 (80.5%)	50 (79.4%)	7 (100.0%)	9 (75.0%)	0.194	0.308	0.412	1	0.213	1	0.43
Number of Lesions (median [range])	5.0 (1–109)	4.5 (1–60)	5.0 (1–109)	5.0 (1–109)	7.0 (6–16)	3.0 (1–20)	0.275	0.186	0.841	0.333	0.617	0.494	0.104
Lines of prior therapy, n (median [range])	2.0 (2–8)	3.0 (2–6)	2.0 (2–8)	2.0 (2–6)	2.0 (2–8)	2.0 (2–5)	0.283	0.156	0.262	0.742	0.589	0.667	0.611
Lines of prior therapy ≥ 3	76 (48.7%)	41 (55.4%)	35 (42.7%)	27 (42.9%)	3 (42.9%)	5 (41.7%)	0.154	0.196	1	1	0.81	0.567	1
Axi-cell CAR T Product	153 (98.1%)	73 (98.6%)	80 (97.6%)	61 (96.8%)	7 (100.0%)	12 (100.0%)	1	0.888	1	1	1	1	1
Leukapheresis to CAR T infusion interval, d (median [range])	27.0 (20–242)	27.0 (20–183)	27.0 (20–242)	27.0 (20–57)	27.0 (21–35)	27.0 (24–242)	0.73	0.18	0.858	0.023	0.613	0.142	0.449

Abbreviations: BT, bridging therapy; ST, systemic therapy; RT, radiation therapy; CMT, combined-modality therapy; ECOG, Eastern Cooperative Oncology Group; PS, performance status; IPI, International Prognostic Index; DLBCL, diffuse large B-cell lymphoma; PMBCL, primary mediastinal B-cell lymphoma; tFL, transformed follicular lymphoma; ABC, activated B-cell-like; GCB, germinal center B-Cell-like; LDH, lactate dehydrogenase; ULN, upper limit of normal; SUV, standardized uptake value; and CAR T, chimeric antigen receptors T-cell.

**Table 2 cancers-15-01747-t002:** Details of patients who received bridging therapy with radiation or combined-modality therapy.

Patient	Comprehensive or Focal RT	# of Sites on PET	RT Dose (Gy)	RT Fractions	RT Target	Concurrent Systemic Therapy	Pattern of Failure
Patient 1	Focal	7	20	5	R neck/mediastinum	R-ICE	Out of field
Patient 2	Comprehensive	1	4	2	Right foot	None	Death without recurrence
Patient 3	Comprehensive	2	4 40	2 20	Left tonsil Head and neck	None	None
Patient 4	Comprehensive	1	14	7	Right lateral chest wall	None	None
Patient 5	Focal	6	30	10	L2-S5	Dexamethasone, intrathecal cytarabine and methotrexate	Out of field
Patient 6	Comprehensive	3	24	12	Head and neck	None	None
Patient 7	Focal	16	29.6 20 20 14	16 7 7 10	Abdomen/pelvis T11 paravertebral One iliac nodes RT humerus	Dexamethasone, pembrolizumab	In field
Patient 8	Focal	7	20	5	Abdomen	R-BENDA-POLA	Out of field
Patient 9	Focal	10	30 20	10 5	Right chest wall Right lower abdomen	R-BENDA-POLA	Out of field
Patient 10	Comprehensive	5	24.75 20	11 10	Right neck Retroperitoneum	None	None
Patient 11	Comprehensive	2	30	10	Bilateral head and neck	None	Death without recurrence
Patient 12	Focal	5	15	5	Left supraclavicular area	None	None
Patient 13	Comprehensive	3	9 7.5 12.5 15 30	3 3 5 5 10	Left thigh Right breast Right breast Right knee Left leg	None	None
Patient 14	Comprehensive	2	30	10	Left neck	None	None
Patient 15	Focal	6	20	10	R foot	None	Out of field
Patient 16	Focal	8	21 20 20	10 10 10	Left neck Right neck Left thigh	R-BENDA-POLA	In field
Patient 17	Focal	20	20 20	10 10	Head and neck axilla	None	Out of field
Patient 18	Focal	6	30	10	Right calf	R-BENDA-POLA	In field
Patient 19	Focal	7	15	5	Sinus	None	Out of field

Abbreviations: RT, radiation therapy; PET, positron emission tomography; R-ICE, rituximab ifosfamide carboplatin etoposide; and R-BENDA-POLA, rituximab bendamustine polatuzumab vedotin.

**Table 3 cancers-15-01747-t003:** Treatment-related toxicity and survival outcomes among patients who received CAR T-cell therapy.

			BT				*p*						
Characteristic	All Patients (n = 156)	No BT (n = 74)	BT (n = 82)	ST (n = 63)	CMT (n = 7)	RT (n = 12)	No BT vs. BT	ST vs. No BT	ST vs. CMT	ST vs. RT	No BT vs. CMT	No BT vs. RT	CMT vs. RT
Best Response							0.51	0.61	0.163	0.663	0.065	0.917	0.139
CR	101 (65.2%)	51 (68.9%)	50 (61.0%)	39 (61.9%)	2 (28.6%)	9 (75.0%)							
PD	15 (9.7%)	6 (8.1%)	9 (11.0%)	6 (9.5%)	2 (28.6%)	1 (8.3%)							
PR	39 (25.2%)	16 (21.6%)	23 (28.0%)	18 (28.6%)	3 (42.9%)	2 (16.7%)							
ORR	140 (90.3%)	67 (90.5%)	73 (89.0%)	52 (82.5%)	5 (71.4%)	8 (66.7%)	0.759	0.441	0.838	0.386	1	0.759	1
CRR	101 (65.2%)	51 (68.9%)	50 (61.0%)	36 (57.1%)	2 (28.6%)	8 (66.7%)	0.322	0.975	0.298	0.769	0.335	0.677	0.259
OS, years							0.158	0.183	0.04	0.38	0.001	0.707	0.021
Median	3.16	NR	3.16	3.16	0.54	3.32							
95% CI	2.09—NR	2.33—NR	1.37—NR	1.33—NR	0.02—NR	1.17–3.32							
PFS, years							0.16	0.176	0.004	0.144	<0.001	0.376	0.001
Median	1.05	1.45	0.65	0.65	0.24	3.32							
95% CI	0.55–1.95	0.58—NR	0.28–1.79	0.28–1.79	0.02–0.45	0.13–3.32							
CRS ≥ 1	133 (85.3%)	58 (78.4%)	75 (91.5%)	58 (92.1%)	5 (71.4%)	12 (100.0%)	0.038	0.048	0.288	0.705	1	0.166	0.237
CRS ≥ 3	11 (7.1%)	5 (6.8%)	6 (7.3%)	3 (4.8%)	1 (14.3%)	2 (16.7%)	1	0.896	0.864	0.377	1	0.551	1
ICANS ≥ 1	68 (43.6%)	31 (41.9%)	37 (45.1%)	28 (44.4%)	4 (57.1%)	5 (41.7%)	0.807	0.898	0.81	1	0.704	1	0.861
ICANS ≥ 3	18 (11.5%)	7 (9.5%)	11 (13.4%)	7 (11.1%)	2 (28.6%)	2 (16.7%)	0.602	0.972	0.475	0.954	0.363	0.804	0.976
Steroid Administration	62 (39.7%)	31 (41.9%)	31 (37.8%)	22 (34.9%)	3 (42.9%)	6 (50.0%)	0.721	0.51	1	0.507	1	0.832	1
Tocilizumab Administration	93 (59.6%)	42 (56.8%)	51 (62.2%)	39 (61.9%)	3 (42.9%)	9 (75.0%)	0.598	0.662	0.569	0.591	0.757	0.381	0.364
ICU Admission	11 (7.1%)	5 (6.8%)	6 (7.3%)	4 (6.3%)	0 (0.0%)	2 (16.7%)	1	1	1	0.531	1	0.551	0.714

Abbreviations: BT, bridging therapy; ST, systemic therapy; RT, radiation therapy; CMT, combined-modality therapy; CR, complete response; PD, progressive disease; PR, partial response; ORR, objective response rate; CRR, complete response rate; OS, overall survival; PFS, progression-free survival; NR, not reached; CI, confidence interval; CRS, cytokine release syndrome; BT, bridging therapy; LDH, lactate dehydrogenase; ULN, upper limit of normal; ICANS, immune effector cell-associated neurotoxicity syndrome; and ICU, intensive care unit.

**Table 4 cancers-15-01747-t004:** Patient and treatment characteristics of patients who underwent CAR T-cell therapy and bridging radiation stratified by extent of radiation volume coverage.

Characteristic	All Patients (n = 19)	Focal (n = 11)	Comprehensive (n = 8)	*p*
Age (median [range])	64.0 (21–75)	66.0 (21–73)	58.0 (41–75)	0.888
>60	12 (63.2%)	8 (72.7%)	4 (50.0%)	0.594
Male	9 (47.4%)	4 (36.4%)	5 (62.5%)	0.508
Race and Ethnicity				0.592
Asian	6 (31.6%)	3 (27.3%)	3 (37.5%)	
Black	1 (5.3%)	1 (9.1%)	0 (0.0%)	
Hispanic White	3 (15.8%)	1 (9.1%)	2 (25.0%)	
Non-Hispanic White	9 (47.4%)	6 (54.5%)	3 (37.5%)	
ECOG PS 2–3	4 (21.1%)	3 (27.3%)	1 (12.5%)	0.834
Stage III/IV	14 (73.7%)	10 (90.9%)	4 (50.0%)	0.141
Extranodal disease	14 (73.7%)	10 (90.9%)	4 (50.0%)	0.141
>1 Site Extranodal disease	11 (57.9%)	10 (90.9%)	1 (12.5%)	0.003
IPI				0.225
IPI ≥ 3	4 (21.1%)	3 (27.3%)	1 (12.5%)	0.834
Pathology				1
DLBCL	16 (84.2%)	9 (81.8%)	7 (87.5%)	
tFL	3 (15.8%)	2 (18.2%)	1 (12.5%)	
Cell of Origin				0.598
ABC	9 (47.4%)	6 (54.5%)	3 (37.5%)	
GCB	7 (36.8%)	3 (27.3%)	4 (50.0%)	
Unknown	3 (15.8%)	2 (18.2%)	1 (12.5%)	
HGBL-DH/TH	4 (21.1%)	3 (27.3%)	1 (12.5%)	0.834
Double Expresser	5 (26.3%)	3 (27.3%)	2 (25.0%)	1
Max Dimension (mm; median [range])	52.0 (13.0–137.0)	66.0 (14.0–137.0)	49.0 (13.0–82.0)	<0.001
Bulky Disease (≥7.5 cm)	6 (31.6%)	5 (45.5%)	1 (12.5%)	0.305
Large Disease (≥5 cm)	10 (52.6%)	6 (54.5%)	4 (50.0%)	1
LDH				0.092
>ULN and <2xULN	5 (26.3%)	1 (9.1%)	4 (50.0%)	
>2xULN	2 (10.5%)	2 (18.2%)	0 (0.0%)	
LDH > 2xULN	2 (10.5%)	2 (18.2%)	0 (0.0%)	0.604
Maximum SUV (median [range])	16.5 (4.8–34.8)	15.7 (4.8–22.1)	23.4 (9.5–34.8)	0.063
Maximum SUV > 10	16 (84.2%)	10 (90.9%)	6 (75.0%)	0.763
Number of Lesions (median [range])	6.0 (1–20)	7.0 (5–20)	2.0 (1–5)	0.002
Lines of prior therapy, n (median [range])	2.0 (2–8)	2.0 (2–8)	2.0 (2–5)	0.964
Lines of prior therapy ≥ 3	8 (42.1%)	5 (45.5%)	3 (37.5%)	1
Axi-cell CAR T product	19 (100.0%)	11 (100.0%)	8 (100%)	1
Leukapheresis to CAR T infusion interval, d (median [range])	27.0 (21–242)	27.0 (21–35)	27.5 (26–242)	0.214

Abbreviations: BT, bridging therapy; ST, systemic therapy; RT, radiation therapy; CMT, combined-modality therapy; ECOG, Eastern Cooperative Oncology Group; PS, performance status; IPI, International Prognostic Index; DLBCL, diffuse large B-cell lymphoma; PMBCL, primary mediastinal B-cell lymphoma; tFL, transformed follicular lymphoma; ABC, activated B-cell-like; GCB, germinal center B-Cell-like; LDH, lactate dehydrogenase; ULN, upper limit of normal; SUV, standardized uptake value; and CAR T, chimeric antigen receptors T-cell.

**Table 5 cancers-15-01747-t005:** Multivariable analysis of factors associated with CAR T-cell toxicity.

Category	OR	−95% CI	+95% CI	*p*
CRS ≥ Grade 1				
Lines of prior therapy, n	0.632	0.432	0.925	0.018
Bridging Therapy				
No BT	Ref			
BT	2.792	1.047	7.45	0.04
>1 Lesions	3.172	1.077	9.349	0.036
CRS ≥ Grade 3				
LDH > 2xULN	5.949	1.32	26.802	0.02
Disease Sites	1.028	1	1.056	0.049
ICANS ≥ Grade 1				
CRS	10.343	2.332	45.88	0.002
ICANS ≥ Grade 3				
LDH > 2xULN	4.643	1.239	17.394	0.023

Abbreviations: OR, odds ratio; CI, confidence interval; CRS, cytokine release syndrome; BT, bridging therapy; LDH, lactate dehydrogenase; ULN, upper limit of normal; and ICANS, immune effector cell-associated neurotoxicity syndrome.

## Data Availability

Research data are stored in an institutional repository and will be shared upon request to the corresponding author.

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
