# Peer review of "Long-Term Follow-Up of Bridging Therapies Prior to CAR T-Cell Therapy for Relapsed/Refractory Large B Cell Lymphoma"

_cancers, 2023, doi:10.3390/cancers15061747_

Round 1

Reviewer 1 Report

In their nicely written paper, Ladbury et al. described the long-term outcomes of various bridging therapies and disease burden prior to CAR T-cell therapy in a large cohort by retrospective analysis of single institution. Given the increasing numbers of CAR T-cell treated patient in routine use, this evaluation is important for common daily practice. As another example, they showed that limited disease with only RT as bridging therapy showed superior outcomes. Although other studies also report on same finding, the results are not completely novel, however additional data which support these findings definitely merit current literature.
Despite the benefit of a single institution analysis with a large number of patients and a long median follow-up, there are important major (and minor) concerns.

Major comments
-
How were the patients selected from a methodological point of view? For example, were all patients who received CART prospectively documented or were they identified using a search of the EHR database?
- Although a large number of patients were included (n=156), only 19 patients received radiotherapy, of which 12 patients received RT alone and 7 in combination with systemic therapy as a bridge. These low numbers of patients included per subgroup make it difficult to draw conclusions. The authors do point out, but the low numbers diminish the impact of the observations. Therefore, the findings and conclusions found will have to be significantly nuanced.
- A large percentage of patients (47.5%) did not receive bridging therapy. This is significantly higher than other studies, explain why this is the case.
-
More specifically, what were criteria for the selection of (no) bridging strategy. For example bridging with RT or without bridging treatment for patients with minimal disease? Besides, what were reasons to choose for focal or comprehensive RT? Without a clear definition, the research population (and results) is extremely biased and conclusion must also be significantly nuanced.
- Additionally, the authors show that patients receiving RT as bridging therapy have the best OS and PFS. Especially patients receiving comprehensive RT. However there is no information reported on disease status/response evaluation after bridging therapy and prior CAR T-cell therapy. Because of this, it is not convincing that the results of the current study support the theory that radiation therapy enhances the CAR T cell response. Please exclude this suggestion.
- Radiation is defined as follow: “Radiation that encompassed all metabolically active tumor at the time of simulation was considered comprehensive, while radiation that only targeted certain sites of disease was considered focal.” This definition is confusing. What criteria were used to choose comprehensive or focal RT? For example, has total tumor volume been taken into account? And what was the effect of RT on tumor volume. If these data is not available, this will definitively impact the outcomes.
-
Considering Table 4/Fig. 2, it appears that patients with focal RT were undertreated as this group had significantly more extranodal sites, larger tumor size and a greater number of lesions compared to patients undergoing extensive RT. The question is whether this is biased by the choice of the doctor. State the criteria on which focal and extended RT are based and nuance the results in this light.
- To better evaluate the results, a multivariable analysis of factors associated with survival outcomes should be included.
- As stated: “In contrast to other published series, patients who received BT did not have inferior OS or PFS that reached statistical significance. This could be attributed to referral patterns at our institution for RT BT, where the threshold for referral is lower and includes bulky or persistent disease that may not be symptomatic or life-threatening, which is likely consistent with more favorable prognosis.” How does the author know his low referral threshold? Have they carried out a comparative study? Please exclude this suggestion and speculate on other reasons based on differences in patient characteristics from other studies, why BT did not have an inferior OS/PFS.

Minor comments
- Simply summarized: evaluate 2x in one sentence
- Please report if all cases are triple FISH for evaluation of BCL2/6/MYC translocation

- There seems to be a bias in the timing of PET scan evaluations. Are all PET scans performed immediately before BT and after bridging? If not, give the reasons and discuss this bias.
- What was the preferred therapy for bulky disease? RT with a boost?
- Were there patients with bulky disease who did not receive bridging therapy? And if so, what were the results?
- In the cohort ‘radiotherapy bridging’ the time between leukapheresis and CAR T infusion varies up to 242 days. How can authors explain this?
- Furthermore, can the authors elaborate on a time correlation between bridging therapy and CART infusion and the effect on the results?
- Kaplan-Meier curves lacking numbers at risk, please add.
- Although informative, Table 1. is far too large (over 3 pages) and not readable in its current form. Please split file for partial presentation of most significant data and include rest data as supplemental.
- One patient received pembrolizumab before CART. What was the response of CART after pembro?
- Oncological results please write out hemato-oncological
-
What measure is taken for Max Dimension (median [range]) (volume or something?)
- As stated in the discussion “The risk of failure appeared to be associated with lesion size as 7/9 failures had a metabolic tumor volume >50 cc.“ How was this measured? When extensive radiological data is used, shouldn't a radiologist be involved in the manuscript? In contrast, sentence 287 “Although metabolic tumor volume estimates are not available in our cohort,”. Is it measured or not, or in part of the cohort?

Author Response

We would like to thank the reviewer for their time and thoughtful comments. Please find our responses below.

  1. How were the patients selected from a methodological point of view? For example, were all patients who received CART prospectively documented or were they identified using a search of the EHR database?

We thank reviewer for this important question. Consecutive patients were prospectively documented in an institutional database. We have added this to the manuscript.

  1. Although a large number of patients were included (n=156), only 19 patients received radiotherapy, of which 12 patients received RT alone and 7 in combination with systemic therapy as a bridge. These low numbers of patients included per subgroup make it difficult to draw conclusions. The authors do point out, but the low numbers diminish the impact of the observations. Therefore, the findings and conclusions found will have to be significantly nuanced.

We agree that there are significant limitations based on sample size, which we discussed in the limitations section. However, to our knowledge our study would still be the largest cohort of bridging RT patients, with the longest follow-up, that has been published. We also hope our findings will increase the awareness of radiation usage as a clinically meaningful bridging therapy.

  1. A large percentage of patients (47.5%) did not receive bridging therapy. This is significantly higher than other studies, explain why this is the case.

We thank the reviewer for this important observation. We would like to clarify that our number is consistent with published series by Saifi et al, where 51/118 (43.2%) did not receive bridging therapy (BT), and Pinnix et al, where 62/124 (50.0%) did not receive BT. Those are the largest series that would be most accurate to compare our series to. There are differences in institutional practice given BT is not standardized. We discuss in the first paragraph of the discussion that we give bRT for bulky/persistent but asymptomatic disease, where other institutions may not.

  1. More specifically, what were criteria for the selection of (no) bridging strategy. For example bridging with RT or without bridging treatment for patients with minimal disease? Besides, what were reasons to choose for focal or comprehensive RT? Without a clear definition, the research population (and results) is extremely biased and conclusion must also be significantly nuanced.

All physicians’ request to treat patients using standard of care CAR T cell products are discussed and approved at our weekly Clinical Cellular Immunotherapy (CCIC) committee meeting. Necessity for BT and types of BT are recommended. But the final determination of BT is at the discretion of the treating hematologist, and where appropriate, radiation oncologist. Comprehensive RT was administered whenever it was felt to be feasible to encompass all disease in the radiation field without excessive toxicity. We have added this clarification to the manuscript in section 2.2. We acknowledge that there is certainly selection bias for each cohort. We included this selection bias in the discussion, and it certainly limits conclusions. However, in our opinion it does not negate our primary conclusions that BT did not appear to compromise outcomes or toxicity as well as the fact that patients with limited disease treated with bridging RT had favorable outcomes.  

  1. Additionally, the authors show that patients receiving RT as bridging therapy have the best OS and PFS. Especially patients receiving comprehensive RT. However there is no information reported on disease status/response evaluation after bridging therapy and prior CAR T-cell therapy. Because of this, it is not convincing that the results of the current study support the theory that radiation therapy enhances the CAR T cell response. Please exclude this suggestion.

Disease status/response is not captured for all patients after BT, and if it is its significance is uncertain, as the intervals between completion of BT and response assessment and CAR T are too short for BT to demonstrate its maximal effect. There is also practical challenge to get payer approval for response scan prior to CAR T cell treatment.  Additionally, BT alone without CAR T cell therapy is unlikely to lead to a durable response based on historical literature and our collective experience.   Our study showed that CAR T treated patients with bRt have the best OS, PFS, and response rates post CAR-T, including relative to no BT, suggest that having two modalities addressing active disease (i.e. RT and CAR T) may be beneficial. We acknowledge due to sample size and selection bias, we cannot make a definitive conclusion on our series. Accordingly, we have deleted “with a potential synergetic impact on outcomes” from our discussion.

  1. Radiation is defined as follow: “Radiation that encompassed all metabolically active tumor at the time of simulation was considered comprehensive, while radiation that only targeted certain sites of disease was considered focal.” This definition is confusing. What criteria were used to choose comprehensive or focal RT? For example, has total tumor volume been taken into account? And what was the effect of RT on tumor volume. If these data is not available, this will definitively impact the outcomes.

We would like to ask please refer to our responses to comment #4 regarding the selection of comprehensive versus focal RT and the practical challenge of conducting disease assessment after BT but prior to CAR T. disease burden is associated with this, which is exemplified in maximum disease diameter and total number of lesions in table 4. Of course this means there is selection bias and confounding, which we write about in the discussion. The best assessment that is available for all patients is disease response after CAR T, where it is not as easy to isolate the effect of BT. However, there is also currently no data on how the effect RT has on tumor burden ultimately has on outcomes.  

  1. Considering Table 4/Fig. 2, it appears that patients with focal RT were undertreated as this group had significantly more extranodal sites, larger tumor size and a greater number of lesions compared to patients undergoing extensive RT. The question is whether this is biased by the choice of the doctor. State the criteria on which focal and extended RT are based and nuance the results in this light.

We appreciate the comment. There are many factors impacting the choice of focal RT versus comprehensive RT for patients. The use of RT in our study is to serve as a bridging therapy to stabilize and or debulk the lymphoma, and get patient to CAR T cell therapy without much delay since CAR T is the treatment with curative intent.   The selection of RT for patients are based on what is clinically feasible and indicated at the time of BT as well as the urgency and timing of CAR T cell availability and initiation which inevitably lead to selection bias. In the discussion section of the manuscript, we include, “Certainly, in both studies this may be driven by selection bias; patients with more limited disease, who already would have better prognosis, are also most amenable to comprehensive RT BT. Nonetheless, treating all gross disease should be considered whenever feasible as a means of improving local control.”   

  1. To better evaluate the results, a multivariable analysis of factors associated with survival outcomes should be included.

We appreciate the suggestion and agree a multivariable analysis of factors associated with outcomes are always desirable. However the small sample sizes of the individual bridging therapy groups in our study make  a MVA not powered to answer questions of interest. We will certainly incorporate this suggestion in our upcoming prospective trial of using RT as bridging therapy.   

  1. “In contrast to other published series, patients who received BT did not have inferior OS or PFS that reached statistical significance. This could be attributed to referral patterns at our institution for RT BT, where the threshold for referral is lower and includes bulky or persistent disease that may not be symptomatic or life-threatening, which is likely consistent with more favorable prognosis.” How does the author know his low referral threshold? Have they carried out a comparative study? Please exclude this suggestion and speculate on other reasons based on differences in patient characteristics from other studies, why BT did not have an inferior OS/PFS.

We appreciate the reviewer’s comment and have removed this from the discussion. We have revised the paragraph to” This could be attributed to the differences in patient population, CAR T choices, timing of initiation of BT including RT, and supportive cares.   

  1. Simply summarized: evaluate 2x in one sentence

We believe this was corrected in the first revision we submitted as we cannot find the referenced sentence in the version returned to is.

  1. Please report if all cases are triple FISH for evaluation of BCL2/6/MYC translocation

We have added to section 2.4 that high-grade lymphoma (double or triple hit) was identified using fluorescence in situ hybridization (FISH)

  1. There seems to be a bias in the timing of PET scan evaluations. Are all PET scans performed immediately before BT and after bridging? If not, give the reasons and discuss this bias.

We would like to clarify, with few exceptions, PET scans are obtained prior to BT and approximately 1 month after CAR T. If a PET scan was obtained after BT but before CAR T it was not used for defining disease characteristics, as the goal of the paper is to assess the combined effect of BT and CAR T. Additionally, all patients included in our study received standard of care CAR T products. It is not a common practice to obtain PET scan after bridging therapy and before lymphodepletion.  

  1. What was the preferred therapy for bulky disease? RT with a boost?

This is a great question. RT was administered at the discretion of the treating radiation oncologist. Based on data that has emerged after the last patient in this series was treated, it is now our preference to treat bulky lesions to a higher dose.

  1. Were there patients with bulky disease who did not receive bridging therapy? And if so, what were the results?

We thank reviewer for this question.  Sixteen patients with disease >=7.5 cm did not receive BT. The outcomes of this small group are as following. The CR rate is 50%, ORR is 87.5%.  

  1. In the cohort ‘radiotherapy bridging’ the time between leukapheresis and CAR T infusion varies up to 242 days. How can authors explain this?

There was a 242 day interval between leukapheresis and CAR T infusion due to development of an active hepatitis B infection that required antiviral treatment before CAR T infusion in one patient RT was given as a BT prior to CAR T. We have added this to the results section.

  1. Furthermore, can the authors elaborate on a time correlation between bridging therapy and CART infusion and the effect on the results?

We thank the review for this great question. The median interval in all groups between leukapheresis and CAR T was 27 days. Shorter interval (< 30 days) did not lead to better 1 year PFS and OS. If you are asking for time between bridging therapy completion and CAR T, that is challenging to define, particularly for systemic therapies that will continue to have action after infusion. It is more clear for RT, but our limited sample size prevents us from drawing conclusions.

  1. Kaplan-Meier curves lacking numbers at risk, please add.

Kaplan-Meier curves have been updated to include numbers at-risk.

  1. Although informative, Table 1. is far too large (over 3 pages) and not readable in its current form. Please split file for partial presentation of most significant data and include rest data as supplemental.

We thank reviewer for the suggestion and have simplified the table to 1 and a half pages and hope this is acceptable.   

  1. One patient received pembrolizumab before CART. What was the response of CART after pembro?

This particular patient rapidly progressed after CAR T despite receiving pembrolizumab. The same patient had 16 sites of disease going into infusion.

  1. Oncological results please write out hemato-oncological

We have changed this heading to Hemato-oncologic outcomes

  1. What measure is taken for Max Dimension (median [range]) (volume or something?)

We thank the reviewer for paying attention to details. The maximal Dimension is measured in mm. We have added the unit to the tables.

  1. As stated in the discussion “The risk of failure appeared to be associated with lesion size as 7/9 failures had a metabolic tumor volume >50 cc.“ How was this measured? When extensive radiological data is used, shouldn't a radiologist be involved in the manuscript? In contrast, sentence 287 “Although metabolic tumor volume estimates are not available in our cohort,”. Is it measured or not, or in part of the cohort?

We would like to point out that the referenced sentence pertains to discussion of work done by Sim et al. (reference 22), not our study. Please refer to the citation immediately preceding that sentence for clarification purpose. We confirm that MTV estimates are not available for our cohort. 

Reviewer 2 Report

I read with interest this manuscript by Dr. Ladbury and colleagues, investigating the outcome of R/R DLBCL patients treated with CART cell therapy. The authors analyzed in particular the efficacy and the safety profile of different bridging therapy solutions, with a main focus on the role of RT.

Several retrospective reports were already published on this topic (properly referenced by the authors), so the novelty is limited. Also, few patients (n=19) received RT as bridging therapy, limiting the evidence on the potential role of RT in this setting.

Despite these limitations (accurately addressed by the author), the manuscript is very well written and reports on a large single-institutional cohort with a significant follow-up time.

I have a few questions for the authors:

·      The RT dose was highly variable (range 4-40 Gy). Despite the mentioned limitations, can the authors address if the RT dose was related with ORR and CR?

·      Considering the literature data, it would be of interest for the readers to know if the authors detect any relapse in field of bridging RT. If yes, which RT dose was used to treat the relapsed lesion(s)? 

·      Patients receiving “comprehensive” bridging RT had a better outcome compared to patients treated with “focal” bridging RT. Anyway, this observation is likely influenced by the different “tumor burden” and disease extension between the two groups. I suggest the authors to better explain this point in the discussion section (actually limited to a brief and generic sentence on the potential selection bias).

·      In table 1 It would be better to replace the “commercial” names with “tisa-cel” and “axi-cel”.

Author Response

We would like to thank the reviewer for their time and thoughtful comments. Please find our responses below.

  1. The RT dose was highly variable (range 4-40 Gy). Despite the mentioned limitations, can the authors address if the RT dose was related with ORR and CR?

We thank the reviewer for this thoughtful comment. Due to sample size it is not possible for us to make meaningful inferences regarding dose. A planned multi-institutional study will optimally be able to better address this question.

  1. Considering the literature data, it would be of interest for the readers to know if the authors detect any relapse in field of bridging RT. If yes, which RT dose was used to treat the relapsed lesion(s)?

This is a great suggestion. We have added a column to Table 2 with this data.

  1. Patients receiving “comprehensive” bridging RT had a better outcome compared to patients treated with “focal” bridging RT. Anyway, this observation is likely influenced by the different “tumor burden” and disease extension between the two groups. I suggest the authors to better explain this point in the discussion section (actually limited to a brief and generic sentence on the potential selection bias).

We appreciate the comment and have clarified that “limited disease” refers to patients with less overall tumor burden and that therefore this finding is expected and represents selection bias.  

  1. In table 1 It would be better to replace the “commercial” names with “tisa-cel” and “axi-cel”.

We absolutely agree. Our intent is to use pharmacologic name whenever possible. We have replaced the commercial names in Table 1 and 4.

Reviewer 3 Report

Thank you for the opportunity to review this study titled ‘Long Term Follow-Up of Bridging Therapies Prior to CAR T-Cell Therapy for Relapsed/Refractory Large B Cell Lymphoma’ by Ladbury et al. This is a very interesting study and I commend the authors on their efforts. With CAR-T cell therapy becoming mainstream in the treatment of LBCL, and with the average delays of 1-2 months in obtain the CAR-T product from time of pheresis, the use of BT is becoming more popular, especially in patients with high-risk disease. However, there is limited guidance available for providers on the BT protocol to use. This is where the results of this study are important. While no major conclusions can be drawn, the key take-aways from this study that I feel carry the most merit are the favorable outcomes with lower dose of radiation after CAR-T suggesting a possible synergistic effect and use of comprehensive radiation being superior to focal. This study is informative and would be interesting for readers of this journal.

Here are a few additional comments:

1.     Since this is a single institution analysis where the protocols of BT may be different from other institutions prior to CAR-T cell therapy, it would be difficult to say if these results can be generalized. The decision about ST and RT type and dosing were left to the discretion of the treating physician. Many different protocols of systemic therapies, radiation therapy and variable dosing may exist at other institutions. I feel that this study done multi-institution would provide a wider variety of treatment protocols including standardized protocols can be included in the analysis and their effects on survival can be assessed.

2.     While there was no significant difference in PFS and OS between the BT and non-BT groups (with a trend towards improved survival in the non-BT group), patients who got BT were more likely to have higher risk disease such as high IPI, and larger tumor burden which could be confounding the result in the favor of the non-BT group in this study as well as previous studies.

3.     RT appears to be superior to systemic therapy, especially comprehensive over focal. While there were significant results in the group that received RT, the patient numbers were small.

4.     I agree with the authors that this data needs to be verified prospectively which should include using standardized BT protocol.

Author Response

We would like to thank the reviewer for their time and thoughtful comments. Please find our responses below.

  1. Since this is a single institution analysis where the protocols of BT may be different from other institutions prior to CAR-T cell therapy, it would be difficult to say if these results can be generalized. The decision about ST and RT type and dosing were left to the discretion of the treating physician. Many different protocols of systemic therapies, radiation therapy and variable dosing may exist at other institutions. I feel that this study done multi-institution would provide a wider variety of treatment protocols including standardized protocols can be included in the analysis and their effects on survival can be assessed.

We agree that a multi-institutional analysis would be better able to answer many questions. One is currently planned, but data are as of yet not available. In the meantime, we feel it is worth reporting our findings given it would be the largest cohort with the longest follow-up to date.

  1. While there was no significant difference in PFS and OS between the BT and non-BT groups (with a trend towards improved survival in the non-BT group), patients who got BT were more likely to have higher risk disease such as high IPI, and larger tumor burden which could be confounding the result in the favor of the non-BT group in this study as well as previous studies.

Indeed, and that explanation has been cited as an explanation for why BT had inferior outcomes in other series. However our series did not observe this trend. We speculate it could be related to different patient population and practice variability among treatment physicians since there is no current consensus on the use of BT.

  1. RT appears to be superior to systemic therapy, especially comprehensive over focal. While there were significant results in the group that received RT, the patient numbers were small.

We agree that small numbers limit conclusions. The findings from our study promote clinical consultation with patients, add value to the body of current knowledge of BT and most importantly, lays the foundation for future prospective studies.

  1. I agree with the authors that this data needs to be verified prospectively which should include using standardized BT protocol.

Thank you. We currently have a prospective protocol pending institutional review board approval, which will optimally be followed by a multi-institutional randomized study.

Round 2

Reviewer 1 Report

the authors have answered the questions thoroughly and adequately, which has significantly improved the quality of the manuscript. I would like to thank the authors for sharing this relevant data.